# CROSS-DOMAIN MICROSCOPY CELL COUNTING BY DISENTANGLED TRANSFER LEARNING

**Zuhui Wang**
State University of New York at Stony Brook
zuwang@cs.stonybrook.edu

## ABSTRACT

Microscopy images from different imaging conditions, organs, and tissues often have numerous cells with various shapes on a range of backgrounds. As a result, designing a deep learning model to count cells in a source domain becomes precarious when transferring them to a new target domain. To address this issue, manual annotation costs are typically the norm when training deep learning-based cell counting models across different domains. In this paper, we propose a cross-domain cell counting approach that requires only weak human annotation efforts. Initially, we implement a cell counting network that disentangles domain-specific knowledge from domain-agnostic knowledge in cell images, where they pertain to the creation of domain style images and cell density maps, respectively. We then devise an image synthesis technique capable of generating massive synthetic images founded on a few target-domain images that have been labeled. Finally, we use a public dataset consisting of synthetic cells as the source domain, where no manual annotation cost is present, to train our cell counting network; subsequently, we transfer only the domain-agnostic knowledge to a new target domain of real cell images. By progressively refining the trained model using synthesized target-domain images and several real annotated ones, our proposed cross-domain cell counting method achieves good performance compared to state-of-the-art techniques that rely on fully annotated training images in the target domain. We evaluated the efficacy of our cross-domain approach on two target domain datasets of actual microscopy cells, demonstrating the feasibility of requiring annotations on only a few images in a new domain.

## 1 INTRODUCTION

Counting cells in microscopy images is useful for many biology discoveries and medical diagnoses (Zimmermann et al., 2003; Trivedi et al., 2015). The number of red blood cells and white blood cells in human bone marrow is a critical indicator of blood cell disorder-related diseases, such as Thalassemia and Lymphoma. Microscopy cell counting is also important in drug discovery for assessing the effects of drugs on cellular proliferation and death. These techniques are employed in cell-based assays to predict drug response. Furthermore, cell count numbers are commonly used as a measure of toxicity in high-content screening for small molecules (Boyd et al., 2020). Counting numbers of crowded cells is a tedious, time-consuming, and error-prone task in real-world applications. Therefore, various deep learning methods have been developed for exact cell counting Cohen et al. (2017); Xie et al. (2018); Guo et al. (2019); Wang & Yin (2021a;b), which is usually achieved by generating a cell density map for a microscopy cell image and then integrating the density map to estimate the total cell number in the image. Three challenges are remaining unsolved in the cell counting problem: (1). **Information entanglement**: a cell image embeds two types of entangled features for cell counting: cell densities in images; and various cell appearances, shapes and various image background contexts related to specific tissues and imaging conditions. Disentangling the mixed features in images and learning informative and discriminative features will facilitate deep learning networks to generate accurate cell density maps while being invariant to specific cells or background contexts; (2). **Costly annotation**: the success of deep learning on cell counting relies on well-annotated training datasets. To relieve the tedious and time-consuming annotation efforts, it is expected to invent weak-annotation cell counting approaches which can learn from a small amount

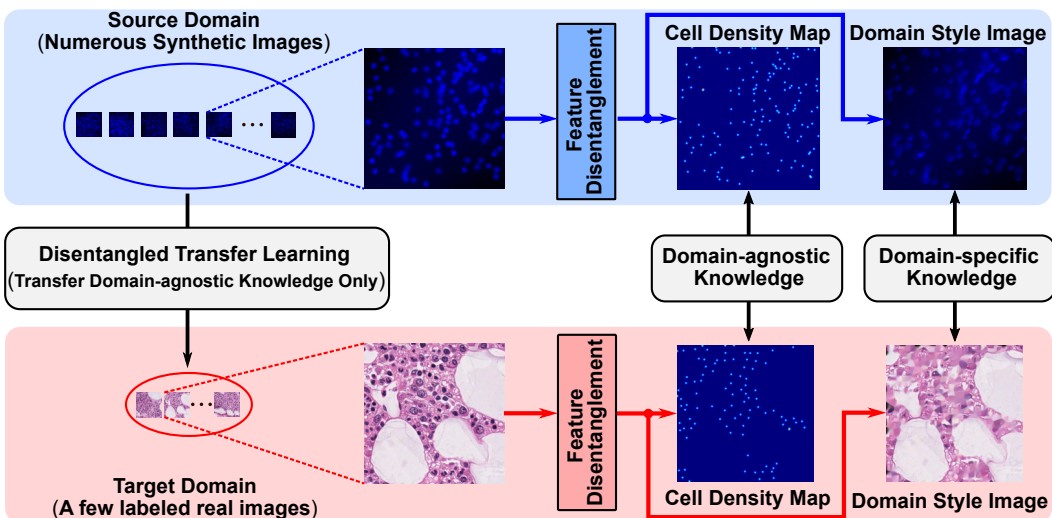

Figure 1: The challenges and our proposal for the cross-domain cell counting task.

of annotated data and use a large amount of synthetic data; (3). **Large cross-domain gaps**. Transferring a model trained on a source domain (e.g., a synthetic dataset) to a new target domain is a promising approach to adapting a cell counting network to different application scenarios. But, when the data distributions between the source and target domains exhibit large variations, the domain gap will hinder the transfer.

## 1.1 OUR OBSERVATIONS AND PROPOSAL.

Firstly, observing the information entanglement challenge, we propose to decouple an input cell image into a *cell density map* corresponding to *domain-agnostic* knowledge and a *domain style image* corresponding to *domain-specific* knowledge, as shown in Figure 1 (right). The domain-agnostic knowledge is related to cell density map generation and remains consistent among various cell counting scenarios, while the domain-specific knowledge related to specific imaging conditions, organs, and tissues is disentangled from the input without affecting the cell counting. Secondly, to reduce the dense annotation cost, we propose to train a cell counting network using a synthetic dataset with known ground truth without manual annotation and then transfer the network to a target domain (Figure 1 (left)). To address the issue of insufficient training data in the target domain, we propose an image synthesis method that can generate a large amount of training data based on a few annotated target-domain images. Thirdly, the data distribution gaps between domains are mainly caused by domain-specific knowledge that is not shared among various domains, so transferring domain-specific knowledge will have a negative impact on adapting cell counting networks across domains. Thus, during transfer learning, we propose to only transfer disentangled domain-agnostic knowledge that is consistent among domains. The contributions of this paper are threefold:

- A novel cell counting network is designed to estimate cell numbers by disentangling input cell images into domain-agnostic knowledge (cell density maps) and domain-specific knowledge (domain style images);

- A new image synthesis method is proposed which synthesizes cell image patches based on a few real images in the target domain and blends a random number of cells into random locations in domain style images, yielding a large number of training images in the target domain for transfer learning;

- A new progressive disentangled transfer learning is proposed for cross-domain cell counting, which effectively transfers the informative and discriminative features learned from a synthetic source-domain dataset to a new target domain with weak annotated images.

Figure 2: Our proposed cell counting network architecture with domain-agnostic and domain-specific knowledge disentangled.

## 2 RELATED WORK

Cell counting methods can be broadly classified into two categories: detection-based methods Arteta et al. (2012; 2016) and regression-based methods (Lempitsky & Zisserman, 2010; Cohen et al., 2017; Xie et al., 2018; Lu et al., 2019). Detection-based methods use a detector to locate cells in images, and the total number of cells is then estimated by counting the detected cells. However, the accuracy of these methods heavily depends on the detector's performance, which can be challenging due to occlusions, various cell shapes, and complex background environments in microscopy images. On the other hand, regression-based methods have recently emerged as state-of-the-art models for cell counting (Lempitsky & Zisserman, 2010). These methods estimate the total number of cells directly using regression models, without the need for explicit detection. Given their promising performance, this paper primarily focuses on discussing and comparing several regression-based cell counting methods.

Most of the state-of-the-art deep learning networks for regression-based cell counting employ density map generation techniques. For instance, Xie et al. (2018) proposed an FCN-based model that generates density maps to estimate cell numbers directly. Similarly, SAU-Net Guo et al. (2019) is an improved model that incorporates a self-attention module into U-Net Ronneberger et al. (2015) to enhance cell counting performance. Another example is Count-ception Cohen et al. (2017), which estimates cell numbers using specially designed cell density maps generated using square kernels. However, these models require domain-specific training data with dense annotations in various domains to achieve promising cell counting results. The availability of a sufficient number of annotated samples is crucial for the success of these state-of-the-art cell counting algorithms.

## 3 METHODOLOGY

In this section, we first introduce a cell counting network which is capable of disentangling domain-agnostic/specific features. Then, we describe how to synthesize target-domain images from a few weak annotated samples. Finally, we present how to transfer a cell counting network trained on a synthetic source domain to a target domain by preserving discriminative information.

### 3.1 CELL COUNTER AWARE OF DOMAIN-AGNOSTIC/SPECIFIC KNOWLEDGE

#### 3.1.1 CELL COUNTER ARCHITECTURE

The proposed cell counter model is illustrated in Figure 2 with four main modules: (1) feature encoder; (2) feature enhancement module; (3) domain-specific decoder; and (4) domain-agnostic decoder. First, for the feature encoder, we employ the first ten layers of a pre-trained VGG16 Simonyan & Zisserman (2015) model, which consists of ten convolutional layers with $3 \times 3$ kernels. After every two or three convolutional layers, the max-pooling layer is applied. The encoder is intended to extract basic feature representations of input images.

Second, the extracted features are then fed into the feature enhancement module that contains an attention submodule and a dilated convolution submodule (Li et al., 2018). The attention submodule consists of spatial and channel-wise attention mechanisms. The spatial attention contains two

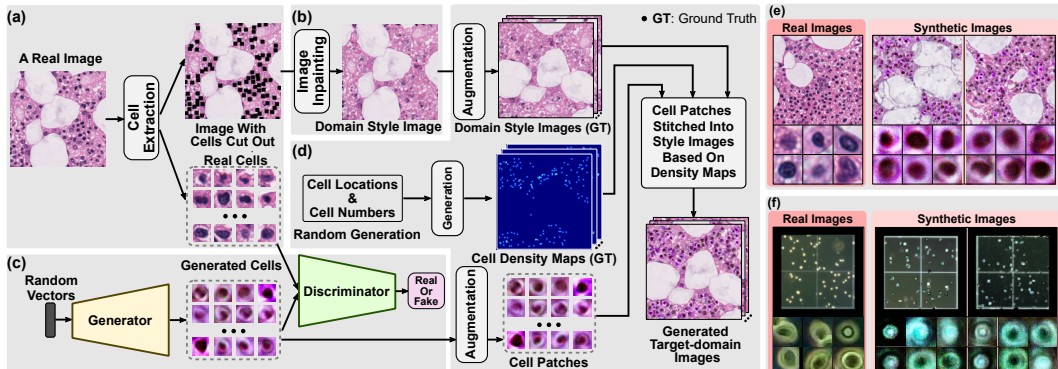

Figure 3: (a)-(d) Our method to synthesize images in a target domain based on a few annotated images. (e)-(f) Some real and synthetic image samples in two target domains.

convolutional layers, with a sigmoid function followed. This spatial attention tends to focus on important regions within encoded features. The channel-wise attention consists of a global average pooling layer and two dense layers followed by a sigmoid function. The channel-wise attention assigns different weights on feature channels to emphasize contributions of various feature maps. The weighted features are then passed to six dilated convolutional layers to extract features at different scales. The dilated convolutional layer is capable to capture crowded target features (i.e., cell features in our project) at multiple-scale environments (Li et al., 2018).

Finally, the enhanced features are sent to the domain-specific decoder and domain-agnostic decoder. Both of these decoders have three up-sampling layers followed by three convolutional layers. The domain-specific decoder attempts to extract features unique to biological experiment domains as domain style images. Simultaneously, the domain-agnostic decoder preserve discriminative features and generates cell density maps for cell number estimation. The domain-specific knowledge, represented by domain style images, varies across domains, so disentangling it out of the input image will enable the transfer learning to correctly transfer the domain-agnostic knowledge shared across domains which control the cell density map generation.

### 3.1.2   LOSS FUNCTION

There are two terms for the total loss function ($L$): Pixel-wise Mean Squared Error ($L_{\mathrm{MSE}}$) loss for the domain-agnostic decoder to generate density maps (i.e., 4th module in Figure 2), and Perceptual loss ($L_{\mathrm{PERC}}$) for the domain-specific decoder (i.e., 3rd module in Figure 2) to measure the high-level perceptual and semantic differences (Johnson et al., 2016). The proposed loss function is written as,

$$L = L_{\mathrm{MSE}} + L_{\mathrm{PERC}} = \frac{1}{N}\sum_{n=1}^{N}\left\|\hat{\mathbf{Y}}_n - \mathbf{Y}_n\right\|_2^2 + \frac{1}{N}\sum_{n=1}^{N}\left\|\hat{\mathbf{\Phi}}_n - \mathbf{\Phi}_n\right\|_2^2, \tag{1}$$

where $\hat{\mathbf{Y}}_n$ is the $n$-th generated density map by the domain-agnostic decoder, $\mathbf{Y}_n$ is the corresponding ground truth cell density map. $\hat{\mathbf{\Phi}}_n$ is the feature map of the $n$-th generated domain style image, $\mathbf{\Phi}_n$ is the corresponding feature map of the ground truth domain style image. $N$ is the total number of samples in the training set.

### 3.2   SYNTHESIZING TARGET-DOMAIN IMAGES BY A FEW ANNOTATED ONES

In this paper, we assume that only a few annotated target-domain images are available, which are insufficient to fine-tune a pre-trained model. Therefore, we consider synthesizing more target-domain images to help a pre-trained model transfer domain-agnostic knowledge across domains. The general workflow of our image synthesis is summarized in Figure 3 with four modules:

- Module-(a) (Figure 3(a)): given an annotated image in the target domain, all cell patches with a resolution of $32 \times 32$ are cropped based on the annotated cell locations, and the remaining image content is regarded as domain-specific knowledge in the target domain;

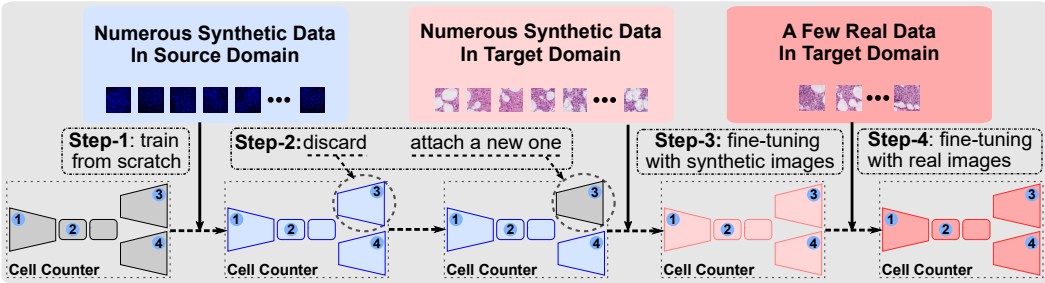

Figure 4: Our progressive disentangled transfer learning workflow.

- Module-(b) (Figure 3(b)): an image inpainting algorithm Bertalmío et al. (2001); Telea (2004) is applied to fill the cut-out cell regions, yielding the domain style image;

- Module-(c) (Figure 3(c)): based on the cropped cells, a Generative Adversarial Network (GAN) Goodfellow et al. (2014) is trained to generate more cell patches. The GAN model consists of four fully-connected layers for the generator and discriminator, respectively;

- Module-(d) (Figure 3(d)): multiple augmentations (e.g., rotation, flip, and scaling) are applied to the domain style image and generated cell patches to increase the data diversity. Then, associated cell density maps are generated using random numbers of cells at random locations. According to the locations in the cell density map, cell patches are stitched into the domain style image.

The generated target-domain images, along with the ground truth of domain style images and their density maps, will be used to transfer a pre-trained model to the new target domain. Figure 3(e)-(f) shows some samples of real and synthesized images and cell patches in two target domains, demonstrating the good quality of our image synthesis and variety of cell samples in target domains. More examples of synthesized target-domain images can be found in Figure 6.

### 3.3 PROGRESSIVE DISENTANGLED TRANSFER LEARNING

The workflow of the proposed progressive transfer learning framework is illustrated in Figure 4. There are four steps to transfer knowledge from the source domain to the target domain: (1). Step-1: cell counter is trained on a source domain of synthetic cells. (2). Step-2: after obtaining the pre-trained model, its domain-specific decoder is replaced with a randomly initialized domain-specific decoder. This is because a domain-specific decoder pre-trained in the source domain contains knowledge specific to the source domain, and keeping it in the following transfer will badly intertwine it with disparate domain-specific knowledge in the target domain. (3). Step-3: the whole model is fine-tuned by the large number of synthesized target-domain images. This step helps the pre-trained model transfer knowledge from the source domain to the target domain. (4). Step-4: the model is further fine-tuned by the small set of annotated real images in the target domain. Through this progressive transfer learning, a cell counter model trained on a source domain of synthetic cells (i.e., no manual annotation cost) is transferred to a target domain of real cell images with only a few annotated ones. Note that, we do not combine the synthesized target-domain images with the annotated ones for joint fine-tuning, because the synthesized set is dominant which makes the joint fine-tuning unbalanced. In fact, the synthesized set bridges the large gap between the source and target domains, and it is a buffer to gradually fine-tune a pre-trained model to a new domain. The comparison between progressive transfer and joint fine-tuning is shown in the ablation study section.

## 4 EXPERIMENTS

### 4.1 DATASETS AND EVALUATION METRICS

(1). **VGG Cell** Lempitsky & Zisserman (2010): this is our source-domain dataset. It is a public dataset of synthetic cells. There are 200 images with a $256 \times 256$ resolution that contain $174 \pm 64$ cells per image. Following the split in Cohen et al. (2017); Xie et al. (2018); Guo et al. (2019), we

Table 1: Experiment results (MAE values) for the two target-domain datasets, $N$ is the number of training samples. (Note: * means the result is absent in the original paper; † stands for the model is directly trained with $N$ target-domain samples, without any transfer learning setting.)

| Methods | MBM ($N = 15$) | | | DCC ($N = 100$) | | | |
|---|---|---|---|---|---|---|---|
| FCRN-A Xie et al. (2018) | 21.3 | | | 6.9 | | | |
| Count-ception Cohen et al. (2017) | 8.8 | | | -* | | | |
| SAU-Net Guo et al. (2019) | **5.7** | | | **3.0** | | | |
| Training scenarios | $N = 2$ | $N = 5$ | $N = 7$ | $N = 2$ | $N = 5$ | $N = 7$ | $N = 10$ |
| Ours | 19.1 | 13.4 | 6.2 | 13.7 | 8.4 | 6.3 | 3.6 |
| Only train with $N$ samples† | 45.3 | 28.4 | 18.1 | 19.6 | 16.5 | 11.0 | 8.6 |

randomly choose 50 images for training samples, 50 images for validation, and the rest for testing. (2). **MBM Cell** Cohen et al. (2017): this dataset contains bone marrow images, which is used as our target-domain dataset. It contains 44 images with a resolution of $600 \times 600$. There are $126 \pm 33$ cells in each image. We follow earlier works Cohen et al. (2017); Guo et al. (2019) by dividing the dataset into three parts: training, validation, and testing, with 15, 14, and 15 images, respectively. (3). **DCC Cell** Marsden et al. (2018): a dataset containing a wide array of tissues and species is used as another target-domain dataset. It has a total of 176 images with a resolution of $960 \times 960$. Each image contains $34 \pm 22$ cells. 80, 20, and 76 images are randomly chosen as the training, validation, and testing datasets, respectively. Mean absolute error (MAE) is employed to evaluate the cell counting model performance, as in (Cohen et al., 2017; Xie et al., 2018; Guo et al., 2019).

## 4.2 EXPERIMENT RESULTS

### 4.2.1 QUANTITATIVE COMPARISONS.

We use the VGG Cell dataset of synthetic cells as the source domain, and test our disentangled transfer learning method on two target domains with real-cell images: MBM Cell and DCC Cell. As shown in Table 1, when using a few annotated images in the target dataset for model transfer (e.g., $N = 7$ in MBM Cell and $N = 10$ in DCC Cell), our method (i.e., *Ours* in Table 1) outperforms two recent methods (FCRA-A Xie et al. (2018) and Count-ception Cohen et al. (2017)) and is comparable to the state-of-the-art (SAU-Net (Guo et al., 2019)). Note that, SAU-Net requires annotating all the target-domain training images for training, but our method only uses a few annotation images in the target domain. In the second comparison, we trained our cell counting network as shown in Figure 2 on a few annotated images in the target domain directly, without any transfer on a pretrained model or synthetic target-domain images. The performance (i.e., *Only train with N samples* in Table 1) of this direct training is much lower than our transfer method.

### 4.2.2 QUALITATIVE EVALUATION.

Figure 5 shows some cell counting examples on the target domains, along with their disentangled cell density maps and domain style images, demonstrating the effectiveness of our disentangled transfer learning. Based on the results of cell density maps, we can observe that generated density maps hold cell locations accurately, and the final cell counting numbers are close to the ground truths in the two public datasets. Moreover, it demonstrates the effectiveness of our proposed disentangled transfer learning by separating domain-agnostic knowledge from input images and preserving discriminative features for the cell number estimation task. On the other side, based on the results of domain style images in Figure 5, we can observe the effective performance of the domain-specific decoder branch in our network. Generated style images capture different cell appearances/shapes and various image background contexts in real cell datasets. Although the generated style images look imperfect (e.g., blurred and dark regions), these generated results still reveal unique cell information in different domains. Moreover, to illustrate the effectiveness of the proposed method (i.e., Section 3.2) for synthesizing target-domain images. More examples of synthesized target-domain cell images are shown in Figure 6. We can observe that the generated synthesized target-domain images look similar to the real images in target domains. This proposed method helps our disentangled transfer learning from a source domain to a new target domain with only a few annotated target-domain samples.

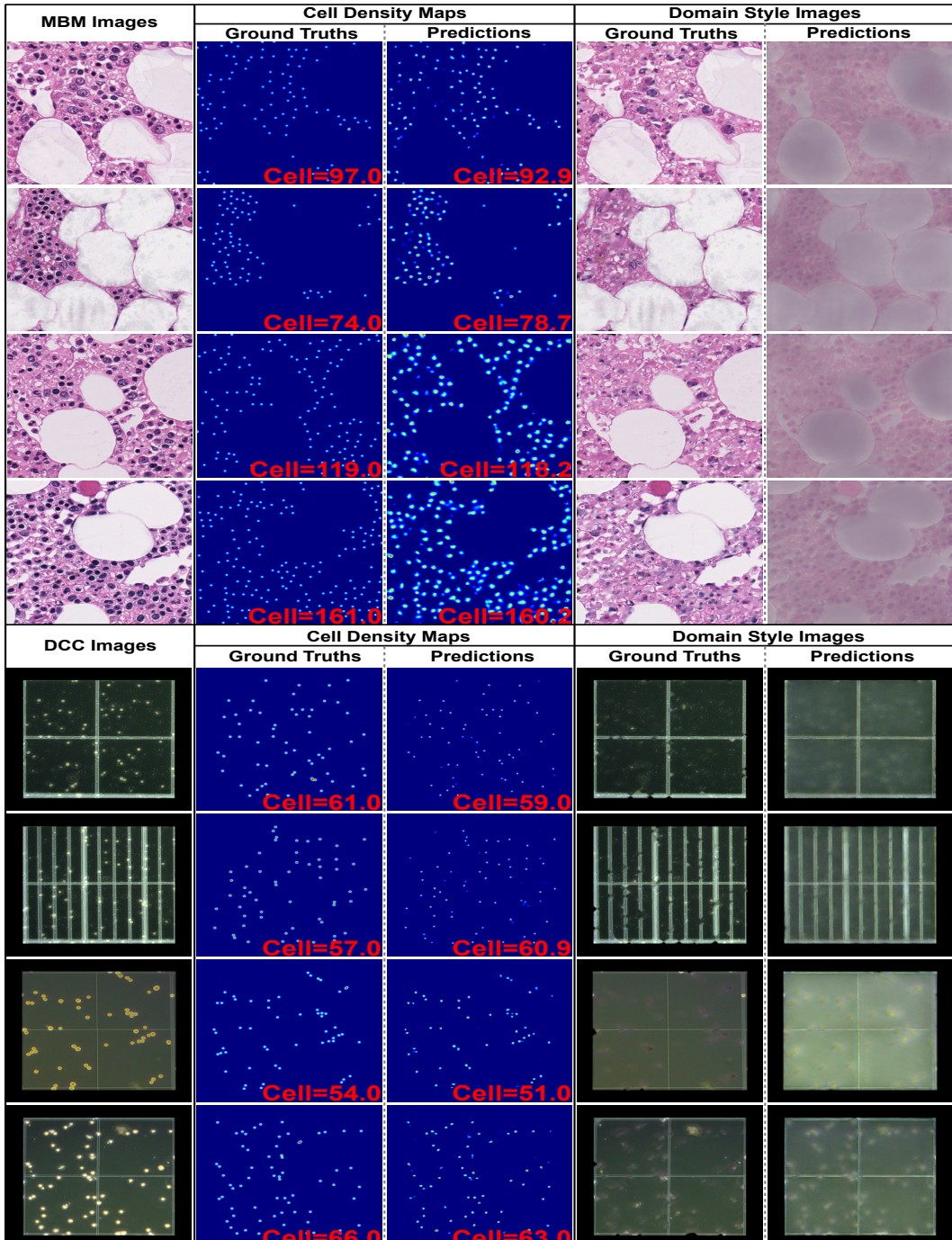

Figure 5: Examples of cell images in the target domains, the disentangled cell density maps, and domain style images compared to ground truth. (Note: cell number in red, and best viewed in color and zoomed in.)

### 4.2.3 ABLATION STUDY

Our proposed method has three key elements: disentangle the domain-agnostic/specific knowledge in cell images; synthesize a large number of images in the target domain; and progressive disentangled transfer a model trained on a synthetic source dataset to a target dataset. We perform three ablation studies (with $N = 7$ in MBM Cell and $N = 10$ in DCC Cell) by removing each element

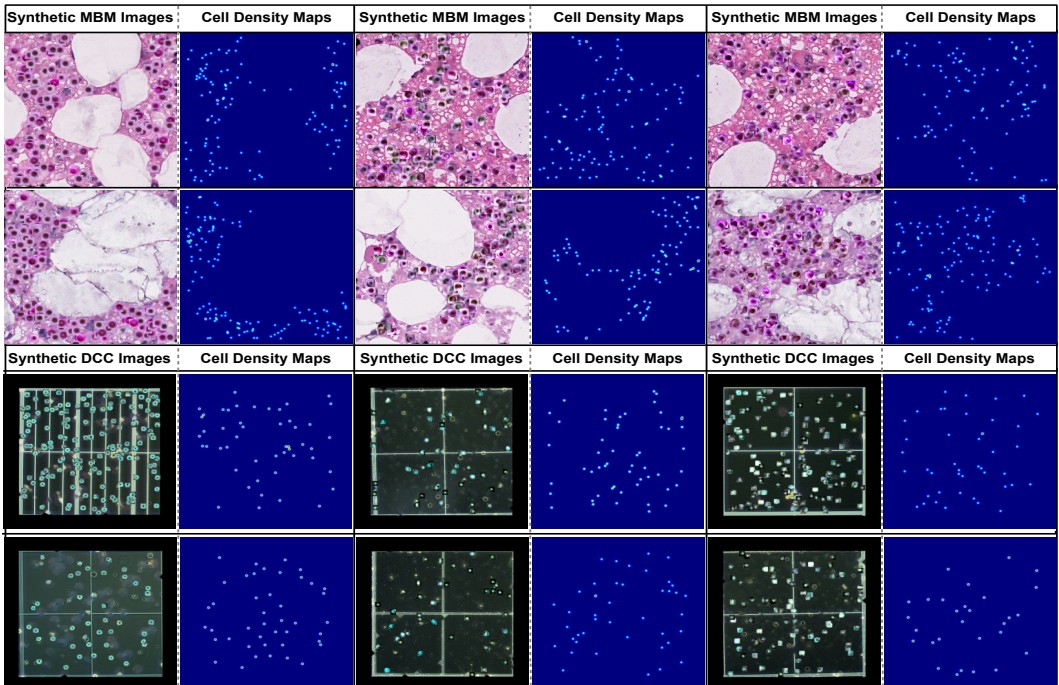

Figure 6: More examples of synthetic cell images and cell density maps in the two target domains.

Table 2: Ablation study for the two target-domain datasets, $N$ is the number of training samples.

| No. | Methods | MBM ($N = 7$) | DCC ($N = 10$) |
| --- | --- | --- | --- |
| 0 | The proposed method (Ours) | 6.2 | 3.6 |
| 1 | Ours w/o disentangling | 13.8 | 5.7 |
| 2 | Ours w/o synthesized target images | 18.8 | 14.9 |
| 3 | Ours w/o progressive fine-tuning | 12.3 | 4.6 |

from our method, as shown in Table 2: No.1 - the disentangling component is removed (i.e., the cell counter network drops the domain-specific decoder); No.2 - the target-domain image synthesis is removed, so the pre-trained model is transferred by a few target-domain images only; and No.3 - the progressive transfer is removed (i.e., the pre-trained model is fine-tuned by a few real images and all synthesized target-domain image at once). We observe that removing any of the three components will lower the model transfer performance, which validates that disentangling domain-agnostic/specific features, synthesizing target-domain images, and progressive transfer, are all helpful in transferring the cell counting network across domains.

## 5 CONCLUSION

Our proposed cell counting model disentangles domain-specific and domain-agnostic knowledge in cell images, allowing for the progressive transfer of only the domain-agnostic knowledge related to cell number estimation from a source domain to a target domain. Leveraging weakly annotated samples in the target domain, our approach preserves discriminative knowledge and effectively discards domain-specific knowledge without negatively impacting the transferring performance. Through evaluation on two real target datasets and one synthetic source dataset, our proposed method achieves good performance in microscopy cell counting than other state-of-the-art methods, while requiring much less annotation effort.

ACKNOWLEDGMENTS

We would like to express our gratitude to the anonymous reviewers for their insightful comments and suggestions, which have significantly enhanced the quality of our paper. We also thank Zhazheng Yin for the computing resource access and discussions.

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
