# OpenReview forum: "Cross-domain Microscopy Cell Counting by Disentangled Transfer Learning"
_ICLR.cc/2023/Workshop/TML4H — ICLR 2023 Workshop TML4H Poster_

### Official Review · Reviewer_N55E · 2023-03-01
**CROSS-DOMAIN MICROSCOPY CELL COUNTING BY DISENTANGLED TRANSFER LEARNING**

**Rating:** 6
**Confidence:** 4

**Review:**

This paper presents a disentangled transfer learning approach for counting cells by annotating only few images.
Pros:
1. It is a novel approach to disentangle input cells into two types of images: domain-agnostic and domain-specific.
2. Uses few real images to generate new images using  a GAN.
3. A novel disentangled transfer learning approach is proposed that transfers informative and discriminative features  from synthetic dataset to a domain of annotated images.

Cons:
1. The paper does not flow well.
2. The paper would have used some more motivation for counting cells.
3. The most important point though is that contrary to the authors claims, generating new images from existing images is not a novelty of this paper. The images are essentially generated using a GAN as specified in Section 3.2 of the paper and there is no novelty in that. It is not clear from the paper description about how this is a novelty of the paper and how it is different from using GANs to generate images.

---

### Meta-Review · Area_Chair_bJYE · 2023-03-05

**Recommendation:** Accept (Poster)
**Confidence:** 5

**Metareview:**

This paper proposes a cross-domain cell counting approach with only weak human annotation efforts.

Strengths:

1. The problem is important in histological image processing.
2. Propose a disentangled transfer learning approach to transferring informative and discriminative features from synthetic dataset to a domain of annotated images.

Weaknesses:

1. The technical contributions are limited.
2. The writing can be further improved.

Overall, I suggest accepting this paper but the authors should carefully consider the comments and further improve the manuscript when preparing the final version.